# Effects of Long-Term Supplementation with Aluminum or Selenium on the Activities of Antioxidant Enzymes in Mouse Brain and Liver

Ilona Sadauskiene [1,*] , Arunas Liekis [1], Inga Staneviciene [2], Rima Naginiene [1] and Leonid Ivanov [2]

[1] Neuroscience Institute, Lithuanian University of Health Sciences, LT-50161 Kaunas, Lithuania; Arunas.Liekis@lsmuni.lt (A.L.); Rima.Naginiene@lsmuni.lt (R.N.)

[2] Department of Biochemistry, Medical Academy, Lithuanian University of Health Sciences, LT-50161 Kaunas, Lithuania; Inga.Staneviciene@lsmuni.lt (I.S.); Leonid.Ivanov@lsmuni.lt (L.I.)

\* Correspondence: Ilona.Sadauskiene@lsmuni.lt; Tel.: +370-(37)-302967; Fax: +370-(37)-302959

**Abstract:** The aim of this study was to investigate the effects of aluminum (Al) or selenium (Se) on the "primary" antioxidant defense system enzymes (superoxide dismutase, catalase, and glutathione reductase) in cells of mouse brain and liver after long-term (8-week) exposure to drinking water supplemented with $AlCl_3$ (50 mg or 100 mg Al/L in drinking water) or $Na_2SeO_3$ (0.2 mg or 0.4 mg Se/L in drinking water). Results have shown that a high dose of Se increased the activities of superoxide dismutase and catalase in mouse brain and liver. Exposure to a low dose of Se resulted in an increase in catalase activity in mouse brain, but did not show any statistically significant changes in superoxide dismutase activity in both organs. Meanwhile, the administration of both doses of Al caused no changes in activities of these enzymes in mouse brain and liver. The greatest sensitivity to the effect of Al or Se was exhibited by glutathione reductase. Exposure to both doses of Al or Se resulted in statistically significant increase in glutathione reductase activity in both brain and liver. It was concluded that 8-week exposure to Se caused a statistically significant increase in superoxide dismutase, catalase and glutathione reductase activities in mouse brain and/or liver, however, these changes were dependent on the used dose. The exposure to both Al doses caused a statistically significant increase only in glutathione reductase activity of both organs.

**Keywords:** superoxide dismutase (SOD); catalase (CAT); glutathione reductase (GR); aluminum (Al); selenium (Se); mouse; brain; liver

## 1. Introduction

Most chemical elements play a very important role in human life. This is especially true for metals, which may have a major impact on human health [1]. Metals are naturally found in the Earth's crust, whereas humans cause their dissemination into the biosphere. These elements are highly stable, water-soluble, can accumulate in soil, and may enter the human body with food, air, or through the skin. Aluminum (Al) is especially noteworthy, as it plays an exceptional role in modern life due to its wide use in both industrial and household contexts. Al compounds are used in water purification and as antacids, food additives, vaccine adjuvants, and antiperspirants. Al intake does not correlate with Al amount in the body [2]. Al absorption by the gastrointestinal tract varies from 0.01% to 1% of the total Al intake. For a long time, owing to its inertness, Al was regarded as a completely harmless metal, but a growing body of emerging evidence suggests that it may be one of the main factors that cause a number of diseases in humans and animals [3]. Due to its specific chemical properties, Al inhibits

more than 200 biologically important functions and causes harmful effects. For instance, chronic exposure to Al has been proven to play a role in the development of neurodegenerative disorders: Parkinson's dementia [4–6] or Alzheimer's disease [7–9]. The mechanisms of Al neurotoxicity are unclear, although research has shown that these mechanisms may mostly be attributed to the ability of Al to: (a) generate reactive oxygen species (ROS) or free radicals when metabolized, and to suppress the activity of antioxidant enzymes and other components of the antioxidant system in various organs, (b) impair signal transduction pathways in the cells, and (c) disturb calcium homeostasis. In addition, Al is one of the causes of hepatotoxicity [10,11] and pathological processes in the testis, kidneys, and lungs [12–14].

However, there is a range of chemical elements that play an important role in the antioxidant processes within cells [15,16]. One such microelement is selenium (Se). The beneficial role of Se in human cells is due to its presence in at least 25 proteins—selenoproteins, part of which are directly involved in redox catalysis. Trace amounts of Se element are essential for cellular functions [17]. Se is an element that reduces the risk of cancer [18], prevents cardiac diseases [19], and protects against the effects of ionizing radiation, heavy metals, and other toxic compounds. There is data indicating that Se strengthens the body's immune system, thus reducing the risk of infection [20]. In large amounts Se and its salts are toxic. Toxic doses of Se are able to negatively affect cellular redox status directly by oxidizing thiols, and indirectly by generating reactive oxygen species (ROS) [21].

Al is the trivalent cation that does not undergo redox changes, but it has a strong prooxidant activity and can potentiate oxidative damages [22]. In addition, high amounts of Se can exert toxic prooxidant properties [21]. The enzymatic and nonenzymatic antioxidant cellular defense systems play a key role in protecting cells from ROS toxicity. Enzymes superoxide dismutase (SOD), catalase (CAT) and glutathione reductase (GR) belong to enzymatic defense system antioxidants. SOD catalyzes the dismutation of two molecules of superoxide anion to molecular oxygen and hydrogen peroxide. Hydrogen peroxide is degraded to water and oxygen by CAT [23]. GR is the essential enzyme for the glutathione redox cycle that maintains the level of most abundant intracellular thiols—glutathione. GR is a nicotinamide adenine dinucleotide phosphate (NADPH)-dependent oxidoreductase, which catalyzes the conversion of oxidized form of glutathione (GSSG) to reduced form (GSH). In its reduced form, glutathione plays key roles in the cellular control of ROS [24].

In our previous experimental studies, we evaluated the effect of acute exposure to Al on oxidative stress and the capacity of the antioxidant system in mouse organs by using the Al intoxication model that involved the injection of $AlCl_3$ solution into the abdominal cavity of the mouse [25–28]. However, the obtained results encouraged us to select a different route of administration of Al: the oral route. This is a natural route of entry of Al into the body, which is also characteristic of humans. In addition, such administration does not cause inflammation at the site of the injection. A number of studies performed with experimental animals have demonstrated changes in the cognitive functions and morphological peculiarities of the CNS following the consumption of water with elevated Al concentrations. Even though the absorption of Al though the gastrointestinal system is very poor, a long-term negative effect of Al cannot be ruled out completely, even if the concentrations of Al that enter the body with potable water are lower. The aim of this study was to evaluate the long-term effect of different doses of Al and Se on the "primary" antioxidant defense system (the enzymes SOD, CAT, and GR) in mouse brain and liver cells after an 8-week oral administration in drinking water supplemented with $AlCl_3$ or $Na_2SeO_3$. This would complement the results of our previous studies on the effects of Al and Se on oxidative stress in mouse tissues.

## 2. Results

Data of the activities of SOD, CAT, and GR in the tissues of control and experimental animals are provided in Tables 1–3. Results have shown that Se affects SOD activity in both brain and liver cells (Table 1). The changes in the activity of the enzyme were observed when the animals received the high dose of Se (0.4 mg/L in drinking water); the activity of this enzyme increased in the brain

(45%) as well as in the liver (33%), compared to controls. Meanwhile, at 8 weeks since the initiation of the experiment, the administration of the Al solution (50 mg or 100 mg Al/L in drinking water) to the laboratory animals had no effect, i.e., SOD activity in the brain and liver of the experiment animals was the same as in the respective organs of the control group animals.

**Table 1.** Superoxide dismutase activity in mouse brain and liver.

| Group | Superoxide Dismutase Activity (U/mg Protein) | |
|---|---|---|
| | Brain | Liver |
| Control | 5.67 ± 1.30 | 0.87 ± 0.10 |
| Al1 | 5.53 ± 0.40 | 0.92 ± 0.24 |
| Al2 | 5.44 ± 0.86 | 0.99 ± 0.17 |
| Se1 | 5.58 ± 1.20 | 0.96 ± 0.16 |
| Se2 | 8.23 ± 0.31 * | 1.16 ± 0.10 * |

The presented data (Mean ± SD) relate to 8–10 experiments. * $p < 0.05$ if compared to the control group. Al1—the mice of the first group; Al2—the mice of the second group; Se1—the mice of the third group; Se2—the mice of the fourth group; C—control group.

**Table 2.** Catalase activity in mouse brain and liver.

| Group | Catalase Activity (U/mg Protein) | |
|---|---|---|
| | Brain | Liver |
| Control | 27.37 ± 6.25 | 963.80 ± 102.04 |
| Al1 | 24.79 ± 6.88 | 944.76 ± 34.90 |
| Al2 | 28.30 ± 9.78 | 955.62 ± 43.22 |
| Se1 | 46.48 ± 17.63 * | 903.27 ± 103.78 |
| Se2 | 41.80 ± 1.96 * | 1121.29 ± 144.25 * |

The presented data (Mean ± SD) relate to 8–10 experiments. * $p < 0.05$ compared to the control group. Al1—the mice of the first group; Al2—the mice of the second group; Se1—the mice of the third group; Se2—the mice of the fourth group; C—control group.

**Table 3.** Glutathione reductase activity in mouse brain and liver.

| Group | Glutathione Reductase Activity (U/mg Protein) | |
|---|---|---|
| | Brain | Liver |
| Control | 9.02 ± 0.89 | 0.88 ± 0.05 |
| Al1 | 11.05 ± 2.17 * | 1.10 ± 0.17 * |
| Al2 | 12.03 ± 1.59 * | 1.18 ± 0.15 * |
| Se1 | 12.28 ± 1.71 * | 0.96 ± 0.07 * |
| Se2 | 11.20 ±1.65 * | 1.18 ± 0.09 * |

The presented data (Mean ± SD) relate to 8–10 experiments. * $p < 0.05$ compared to the control group. Al1—the mice of the first group; Al2—the mice of the second group; Se1—the mice of the third group; Se2—the mice of the fourth group; C—control group.

Similar results (by trend) were obtained when analyzing CAT activity in brain and liver cells of experimental animals. Like in the case of SOD, evident changes in CAT activity were observed in mice that received supplementation with the Se solution for 8 weeks (Table 2): CAT activity was statistically significantly increased in the brain (53%) and liver (16%) of mice that were administered the high dose of Se (0.4 mg/L in drinking water), compared to that observed in the brain and liver of the control mice. However, it is noteworthy that during our experiment, the change in brain CAT activity was also observed in mice that received low doses of Se (0.2 mg/L in drinking water)—in this case, CAT activity increased up to 69%, compared to the control mice. The administration of both doses of Al caused no changes in activity of this enzyme in mouse brain and liver.

The results of the experiment showed that the greatest sensitivity to the effect of the Al and Se was exhibited by GR, and the effect of Al and Se was observed in both the brain and the liver. In

addition, an increase in GR activity was registered at all the doses of Al or Se salts administered to the experimental animals (Table 3). Thus, following the administration of Al, GR activity in mouse brain cells increased by 24% and 34% (at the Al dosage of, respectively, 50 mg and 100 mg/L in drinking water), in comparison to control. Se also increased GR activity in the brain by 36% (0.2 mg/L in drinking water) and 24% (0.4 mg/L in drinking water), respectively. The increase in mouse liver GR activity was dose-dependent: 10% and 33% in 0.2 mg/L and 0.4 mg/L drinking water, respectively. Similar changes were observed in the liver GR activity after exposure to Al solutions. The administration of both low and high doses of Al caused a statistically significant increase in this enzyme activity by 22% and 33%, respectively.

## 3. Discussion

As we have mentioned before, the toxicity of chemical elements (both metals and non-metals) is believed to manifest itself mainly through the formation of ROS or free radicals in the cells of living organisms or through the inhibition of the enzymes that are a part of the antioxidant system. Thus, the cells enter a state that is known as oxidative stress. This stress reflects an imbalance between ROS formation and the ability of the cell's biological system to neutralize these radicals. The antioxidants molecules that constitute the antioxidant defense system act at three different levels: Prevent from radical, scavenge radical and repair radical induced damage [29]. Under usual conditions, a certain pro-oxidant/antioxidant balance is maintained in the cell, thus protecting the cell (its macromolecules) against the destructive effect of the ROS. However, due to various internal and external factors (disease or unfavorable environmental conditions, etc.) this balance may shift in favor of pro-oxidants [30]. Following the depletion of the antioxidant system resources, an uncontrollable increase in ROS in the cell causes oxidative damage to the essential structures of the cell—lipids, nucleic acids, and proteins [31]. Thus, by damaging proteins, nucleic acids and lipids in the cell, ROS plays a key role in the etiology of a number of diseases. In the human organism, oxidative stress is believed to be an important factor in the development of cancer [32], Parkinson's and Alzheimer's diseases, myocardial infarction, heart failure and atherosclerosis [33], the X syndrome [34], autism [35], infection [36], the chronic fatigue syndrome [37], and other diseases.

To ensure protection against oxidative stress, the cells have the so-called antioxidant system. This system is as varied as the radicals that cause oxidative stress. Despite the significant progress in the understanding of the activity of individual enzymes and components of the antioxidant system, the complex composition of the cell's antioxidant system greatly complicates the understanding of the overall functioning of this "defense system". Even though there are several classifications of the antioxidant system in the cells, the classification proposed by Davies [38] in 1988 may be considered one of the most acceptable ones. According to this classification, the antioxidant system is classified into the primary and the secondary ones. The primary antioxidant "defense" system consists of the enzymes such as SOD, CAT, GR and the antioxidant vitamins E, A, C, glutathione, and uric acid. SOD are enzymes that catalyze the dismutation of the superoxide radical, which causes many types of cell damages, into molecular oxygen or hydrogen peroxide. Hydrogen peroxide is also damaging and has degraded by other enzymes such as CAT. GR catalyzes the NADPH-dependent reduction of oxidized glutathione (GSSG) to reduced glutathione (GSH), which plays an important role in the GSH redox cycle that maintains adequate levels of reduced glutathione. The GSH/GSSG ratio determines cell redox status of cells. At rest state healthy cells have a GSH/GSSG ratio greater than 100. When cells are in oxidative stress conditions, the GSH/GSSG ratio drops from 1 to 10 [39]. Glutathione directly scavenges free radicals or is the electron donor for the reduction of peroxides in antioxidant enzymes reactions [40]. Glutathione also is as a thiol buffer maintaining sulfhydryl groups of many proteins in their reduced form.

The secondary antioxidant system includes lipid-cleaving enzymes—phospholipases, proteolytic enzymes—proteases and peptidases, DNA-repairing enzymes—polymerases, glycosylases, endonucleases, exonucleases, and ligases. Concerning the antioxidant "defense" system, of special

interest is the enzyme group of this system—superoxide dismutase, catalase, and glutathione peroxidase/glutathione reductase system. They are the elements that form the "first line of defense" against the ROS and free radicals. The increased GR activity improves the cells ability to replenish glutathione, and helps maintain favourable GSH/GSSG ratio during oxidative stress.

As we have mentioned before, the aim of this study was to evaluate the effect of 8-week oral administration of Al and Se salts on the activity of the main enzymes of the antioxidant defense system (SOD, CAT, and GR) in mouse brain and liver. It is noteworthy that our interest in the study on the effect of Al and Se on the elements of the cellular antioxidant system has also been stimulated by the fact that there are a number of studies on this issue, but they use a different animal intoxication scheme [41].

First, we evaluated the activity of the studied enzymes in mouse brain cells. The evaluation showed that after 8-week oral administration with $AlCl_3$ and $Na_2SeO_3$ solutions, Se ions had the greatest influence on SOD, CAT, and GR activity (by increasing it). During our experiment, the most significant increase in the activity of these enzymes was observed with the administration of the high dose of Se (0.4 mg Se/L in drinking water) by 45%, 53%, and 24%, respectively. Se at low doses (0.2 mg Se/L in drinking water) affected only CAT and GR activity (it increased by, accordingly, 69% and 36%). However, Al (both doses of Al; 50 mg and 100 mg Al/L in drinking water) had virtually no effect on SOD or CAT activity. Only GR proved to be sensitive to the exposure of the experimental mice to both doses of Al. Al ions increased the GR activity by up to 23% and 33%, respectively.

During the second stage of our study, we evaluated the effect of Al and Se on the activity of the antioxidant system enzymes (SOD, CAT, and GR) in hepatocytes. The results of the study showed that the effect of the aforementioned metal salts after 8-week daily oral supplementation caused an increase in SOD, CAT, and GR activity. The effect was analogous to that observed in the brain cells of the same experimental animals. Only a high dose of Se affected SOD and CAT activity (the increase was, accordingly, 33% and 16%), and only GR reacted to the administration of both Al and Se (at all doses). The administration of both low and high doses of Al caused a statistically significant increase in this enzyme activity by 22% and 33%, respectively. Se also increased GR activity in the liver by 10% (low dose) and 33% (high dose), respectively.

It is noteworthy that, compared to the literature data [42–47], our findings partially correlate with those of other studies. For instance, the results of our previous studies showed that exposure to Al tends to reduce the activity of the main enzymes of the antioxidant system [25,26]. Such a trend was observed in both the brain and the liver of laboratory animals [42–45]. Meanwhile, our results showed that the effect of Al on SOD and CAT activity (in brain and liver) was not as significant. GR was an exception here, as its activity statistically significantly increased in both the brain and the liver. Conversely, exposure to Se results in an increase in the activity of all the enzymes of the antioxidant system. In addition, the literature sources have emphasized the antioxidant effect of Se in neutralizing the oxidative stress caused by Al (as well as other factors) [42,43,46,47].

Thus, the results of our study showed that elements (in this case, Al and Se) did affect the components of the antioxidant system in cells of some organs. As expected, the effect depended on the type of the elements. The much weaker effect of Al on some enzymes (SOD and CAT) might have been dependent of its physical and chemical properties; this metal has a relatively difficult entry into the body and very slowly accumulates in organ cells [48]. Al is poorly absorbed in the gastrointestinal tract (0.1%–1.0% of the oral dose) [49]. As a result, symptoms of oxidative stress develop much more slowly. It is known that Al circulates in the blood mainly bound to transferrin (90%) and low molecular mass compounds (e.g., citrate) [50]. Therefore, Al may interfere with Fe homeostasis by displacing it from transferrin; as a result, Fe is released into the bloodstream. Nayak [51] indicated that exposure to Al can impair intestinal Fe absorption. Al ions increase Fe concentration in serum and disrupt normal ferritin level. Al increases uptake of transferrin bound Fe and the transport of non-bound Fe in human glial cells. The higher levels of intracellular Fe can increase oxidative damages [52]. Oxidative damages of cellular lipids, proteins and DNA and disturbed redox status in cells can been

characterized by the change in antioxidant enzymes activities and the consumption of sulfhydryl measured by the reduced glutathione (GSH) status [53]. Glutathione reductase is an enzyme which causes a reduction of the oxidized glutathione. Reduced glutathione is a component of the antioxidant system that comprises more than 90% of total intracellular thiols. It is possible that in our experiments using different doses of aluminum (50 mg or 100 mg Al/L in drinking water), Al enters the brain poorly, and so we did not determine the effect of Al ions on the activities of antioxidant enzymes SOD and CAT. It has been found that only 1% of the total body Al accumulates in brain tissue [54]. On the other hand, it is also known that Al increases the uptake of Fe to brain cells. Fe, a redox-active metal, can interact with molecular oxygen to form the superoxide anion, which in turn generates a highly reactive hydroxyl radical [49]. The role of the intracellular antioxidant GSH is important against these radicals. Reduced GSH can be oxidized to glutathione disulfide (GSSG) during oxidative stress. Reduced GSH is regenerated from GSSG during the glutathione redox cycle catalyzed by GR; therefore, we found an increase in the activity of this enzyme in mice brains after long-term (8-week) exposure to drinking water supplemented with AlCl3 (50 mg or 100 mg Al/L in drinking water). We also found similar changes in GR activity in both the liver and the brain.

Se is important for a various biological process in mammalian cells [55]. Its beneficial role is related to low molecular weight Se compounds, as well as to its presence in at least 25 selenoproteins. Some selenoproteins are directly involved in redox catalysis. Se as a cofactor of glutathione peroxidase increases the antioxidant capacity of intracellular systems [56]. The amount of Se in the human brain is low (2.3%) [57], however, the brain uptake of Se has a high preference in the presence of Se dietary deficiency. The decrease of blood Se concentration and normal brain Se concentration was determinate in young rats after Se-deficient 13-week diet [58]. The deficient supply of Se may have a harmful effect on brain cells, can disturb neuronal function and induce cell death [59]. High metabolic activity of the brain causes excessive production of ROS. The nervous tissue is abundant in Fe, a substrate for the Fenton reaction, and is rich in polyunsaturated fatty acids, which are the target for lipid peroxidation. For these reasons, the brain needs highly efficient ROS scavenging that is mainly mediated by Se-required enzymes [60]. Thus, not surprisingly, exposure to this element significantly affects the enzymes of the antioxidant system. However, it is also known, that Se seems to have both beneficial and harmful properties. This element has a two-sided effect depending on its concentration. The toxic doses of Se are able to negatively affect cellular redox status (directly by oxidizing thiols, and indirectly by generating reactive oxygen species) [21]. It was shown that neurotoxicity of inorganic Se compounds is higher in comparison to organic Se compounds [61], with some exceptions, like methylseleninic acid, selenodiglutathione, L-selenocystine. In our experiments, two doses of Se were chosen (0.2 mg or 0.4 mg Se/L in drinking water), but the effect of both doses of Se on the activity of antioxidant enzymes was similar. The data obtained by us are not sufficient to determine whether the effect of Se on the first line of defense is positive, leading to an increase in the activities of CAT, SOD and GR enzymes. Se may act as prooxidant, forcing cells to increase their antioxidant defenses.

Thus, considering the obtained results, we think it is expedient for further study to assess the effect of metals (especially Al) and microelements (Se) on the components of the antioxidant system in organ cells, paying special attention to the distribution of the studied elements in mice organs.

## 4. Materials and Methods

### 4.1. Materials

All the reagents and solvents used through experiments were of analytical grade (>99.9%). Nitroblue tetrazolium and tris(hydroxymethyl)aminomethane were purchased from Sigma-Aldrich GmbH (Buchs, Switzerland); aluminum chloride hexahydrate and phenazine methosulfate were obtained from Sigma-Aldrich (Steinheim, Germany). All other chemicals were from Sigma-Aldrich (St. Louis, MO, USA).

### 4.2. The Object of Research

The experiments were conducted on 4–6-week-old BALB/c laboratory mice (male) weighing 20–25 g. All experimental procedures were performed according to the Law of the Republic of Lithuania Animal Welfare and Protection (License of the State Food and Veterinary Service for working with laboratory animals No. G2-80). The mice were maintained and handled at Lithuanian University of Health Sciences animal house (Kaunas, Lithuania) in agreement with the ARRIVE guidelines.

### 4.3. The Model of Mouse Intoxication

The mice were given drinking water supplemented with metal salts on a daily basis for 8 weeks. The mice of the first group (Al1) were given drinking water supplemented with $AlCl_3$ (50 mg (1.85 mmol) of Al/L in drinking water). The mice of the second group (Al2) were given drinking water supplemented with $AlCl_3$ (100 mg (3.7 mmol) of Al/L in drinking water). The mice of the third group (Se1) were given drinking water supplemented with $Na_2SeO_3$ (0.2 mg (2.5 μmol) of Se/L in drinking water). The mice of the fourth group (Se2) were given drinking water supplemented with $Na_2SeO_3$ (0.4 mg (5.0 μmol) of Se/L in drinking water). The control group mice had free access to drinking water without supplements. Each experimental and control group included 8–10 mice.

After the exposure time, the animals were terminated according to the rules defined by the European Convention for the Protection of Vertebrate Animals Used for Experimental and Other Purposes.

### 4.4. The Brain and Liver Homogenates Preparation

Following cervical dislocation of the animal the brain and liver were removed, washed, and immediately cooled on ice. The organs were carefully weighed and homogenized in three volumes (relative to organ weight) of buffer (50 mM Tris-HCl, pH 7.6; 250 mM sucrose; 60 mM KCl; 5 mM $MgCl_2$; 10 mM 2-mercaptoethanol). Homogenate was centrifuged at 15,000× *g* for 15 min (centrifuge „Beckman J2-21", USA), and then postmitochondrial supernatant was used for the measurement of enzymatic activity in organ tissues.

### 4.5. Protein Concentration Assay

Protein concentration in homogenate samples of the brain and liver was measured by using the Lowry method [62].

### 4.6. SOD Activity Assay

SOD activity in brain and liver homogenates was evaluated according to the method described in [63]. Activity of enzyme was evaluated spectrofotometrically (spectrophotometer LAMBDA 25, USA) by the inhibition of nitroblue tetrazolium (NBT) recovery rate in the nonenzymatic system with phenazine methosulfate and NADH (nicotinamide adenine dinucleotide/reduced) at 540 nm light wavelength. The SOD activity was expressed as U/mg protein per 1 min, where U was a relative unit of activity defined as the amount of SOD required for the inhibition of NBT reduction by 50% and expressed as a unit of activity in a 1 mg protein sample.

### 4.7. CAT Activity Assay

CAT activity in brain and liver homogenates was evaluated according to the method described in [63]. CAT activity is measured by hydrogen peroxide reaction with ammonium molybdate which produces a complex that absorbs at 410 nm light wavelength. The results were expressed in U/mg protein. Under these conditions, one unit of catalase (U) decomposes 1 μmol of hydrogen peroxide per 1 min.

*4.8. GR Activity Assay*

GR activity in the homogenates was evaluated according to the method described in [64]. The GR reaction rate was evaluated according to the reduction in the optical density in the presence of NADPH (nicotinamide adenine dinucleotide phosphate/reduced) oxidation reaction that occurs during glutathione reduction catalyzed by GR. The GR activity was evaluated spectrophotometrically spectrofotometrically (spectrophotometer LAMBDA 25, USA) at 340 nm wavelength. The results were expressed in U/mg protein. The GR activity unit (U) was the amount of the enzyme catalyzing the formation of 1 μmol of the reaction product in 1 min.

*4.9. Statistical Analysis*

The data were expressed as mean ± SD (standard deviation). Because the mice were grouped into experimental groups according to one factor, statistical significance was assessed using one-way analysis of variance (ANOVA) followed by Tukey's test with the software SPSS Statistics 20.0 (IBM Corporation, NY, USA). The value of $p < 0.05$ was regarded as statistically significant.

## 5. Conclusions

Se is an element that can increase the capacity of the intracellular antioxidant system; Al might have a lesser effect. Exposure to drinking water supplemented with $Na_2SeO_3$ (0.2 mg or 0.4 mg Se/L in drinking water) resulted in an increase in antioxidant potential in mouse brain and liver through enhancing the activities of superoxide dismutase, catalase and especially glutathione reductase, which plays a pivotal role in maintaining the redox state of the cell. Exposure to both doses of $AlCl_3$ (50 mg or 100 mg Al/L in drinking water) resulted in a statistically significant increase in brain and liver glutathione reductase activity but caused no changes in the activities of SOD and CAT.

**Author Contributions:** Data curation, I.S. (Ilona Sadauskiene) and A.L.; Formal analysis, A.L.; Methodology, I.S. (Inga Staneviciene); Project administration, L.I.; Supervision, L.I.; Writing—Original draft, A.L. and R.N.; Writing—review and editing, I.S. (Ilona Sadauskiene) and I.S. (Inga Staneviciene). All authors have read and agreed to the published version of the manuscript.

**Funding:** This research received no external funding.

**Conflicts of Interest:** The authors of this study declare that they have no conflict of interest.

**Data Availability:** The data used to support the findings of this study are available from the corresponding author upon request.

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
