# Peer review of "Effects of Long-Term Supplementation with Aluminum or Selenium on the Activities of Antioxidant Enzymes in Mouse Brain and Liver"

_catalysts, doi:10.3390/catal10050585_

Round 1

Reviewer 1 Report

I agree with the response from author.

Author Response

Thank you very much for your time spent for the evaluation of our manuscript.

Reviewer 2 Report

This paper deals with the activity of the some enzymes in two organs under Al or Se dose. The paper can be published after the following problems are corrected.

  • In title, the authors write the paper deals with enzyme status. However, the authors measured the enzyme activity. How the authors think about? Does the measurement of enzyme activity lead the status evaluation?
  • The time courses of enzymes activity were not shown. How is this?
  • For the enzymes activities of brain catalase of Se1, liver catalase of Se2, brain GRs of Al1, Al2, Se1 and Se2, based on the error magnitude and control value, it would be difficult to say that the enzyme activity is enhanced by the doses. How the authors think about?
  • What the increase of the enzymes activity lead? What conclusion is obtained from the results? The authors should discuss these points.

Author Response

  1. In title, the authors write the paper deals with enzyme status. However, the authors measured the enzyme activity. How the authors think about? Does the measurement of enzyme activity lead the status evaluation?

Thank you for your comment. Maybe in fact the title does not quite correspond to the essence of the research. So according to your suggestion, we change the title of our article into “Effects of long-term supplementation with aluminum or selenium on the activities of antioxidant enzymes in mouse brain and liver”.

  1. The time courses of enzymes activity were not shown. How is this?

Enzymes activities meanings are described in chapter “Materials and methods”. All of them are expressed as U/1 mg protein in 1 min.

  1. For the enzymes activities of brain catalase of Se1, liver catalase of Se2, brain GRs of Al1, Al2, Se1 and Se2, based on the error magnitude and control value, it would be difficult to say that the enzyme activity is enhanced by the doses. How the authors think about?
  2. What the increase of the enzymes activity lead? What conclusion is obtained from the results? The authors should discuss these points.

According to the literature date much weaker effect of Al on some enzymes (SOD and CAT) might have been depended of its physicals and chemicals properties, this metal has a relatively difficult entry into the body and very slowly accumulates in organ cells (reference No 38 in our manuscript). Al is poorly absorbed in the gastrointestinal tract (0.1-1.0% of the oral dose) (39, 44) As a result, symptoms of oxidative stress develop much more slowly. So, we chose different doses to evaluate the effect of the elements used in the experiment and not to observe the death of the animals from an overdose. Maybe that in our experiments using selected doses of aluminum (50 mg or 100 mg Al/L of drinking water), Al enters the brain poorly, so we did not determine the effect of Al ions on the activities of antioxidant enzymes SOD and CAT.

It is known, that only 1% of the total body Al accumulates in brain tissue (44). But it is also known that Al increase the uptake of Fe to brain cells. Fe, a redox-active metal, can interact with molecular oxygen to form the superoxide anion, which in turn generates a highly reactive hydroxyl radical (39). The role of the intracellular antioxidant GSH is important against these radicals. Reduced GSH can be oxidized to glutathione disulfide (GSSG) during oxidative stress. Reduced GSH is regenerated from GSSG during the glutathione redox cycle catalyzed by GR, therefore, we found an increase in the activity of this enzyme in mice brain after long-term (8-week) exposure to drinking water supplemented with AlCl3 (50 mg or 100 mg Al/L of drinking water).

It is also known, that Se seems to have both beneficial and harmful properties. This element has a two-sided effect depending on its concentration. The toxic doses of Se are able to negatively affect cellular redox status (directly by oxidizing thiols, and indirectly by generating reactive oxygen species) (51). It was shown that neurotoxicity of inorganic Se compounds is higher in comparison to organic Se compounds (52), with some exceptions, like methylseleninic acid, selenodiglutathione, L-selenocystine. In our experiments, two doses of Se were chosen (0.2 mg or 0.4 mg Se/L of drinking water), but the effect of both doses of Se on the activity of antioxidant enzymes was similar. The data obtained by us are not sufficient to say whether the effect of Se on the first line of defense is positive, leading to an increase in the activities of CAT, SOD and GR enzymes. Perhaps, Se acts as prooxidant, forcing cells to increase their antioxidant defenses. Thus, considering the obtained results, we think it is expedient to further study to assess the effect of metals (especially Al) and microelements (Se) on the components of the antioxidant system in organ cells, paying special attention to the distribution of the studied elements in mice organs.

Round 2

Reviewer 2 Report

As the first comments, the evaluation of the enzyme activity increase would be insufficient for some data. The authors should consider the error to the values measured. Especially, the authors should write how the increase phenomena was evaluated. The sentence of “Time course of enzyme…” in the first comment means the enzyme activity during the metal doses. It seems that it is difficult to measure the enzyme activity every day. The authors measured the activities after 8 weeks, however, it is not clear that the dosing period is appropriate or not. If the authors have the data on the enzyme activity during the dose period, it would be clear that the measurement condition (= dose 8 weeks) would be preferred. The authors should also write how the dose period and amount are decided. It would be better to be careful about the digit number of significant figures. It is wonder that the digit number is 4 or 5 for the measurement results.

Author Response

Thank you for your comments and questions.

In Laboratory of Molecular Neurobiology of Neuroscience Institute LUHS we use several experimental models of metals impact on mice administered intraperitoneally or per os. We measure enzyme activity in vitro after some certain period but nor during the period. The dosage is chosen while evaluating the LD50, the metals impact is measured after 2, 8, 16 or 24 hours (acute intoxication), 14 days (sub-acute intoxication) or 8 weeks (sub-chronic intoxication). In this research model the dosage and duration were chosen based on literature data, mice got concentration of Al or Se salts solutions with drinking water for 8 weeks in doses described in chapter 4.2. The model of mouse intoxication. (lines 267-278). The enzyme activity was measured after 8 weeks in homogenized organs and not blood. Thus, we didn’t have any possibilities to monitor the enzyme activity during the experiment. Moreover, the aim of our research was to evaluate the antioxidant properties of certain enzymes after long-term Al/Se impact. Changes in enzyme activity after Al and Se impact were compared to the control group of mice administered only drinking water in situ.

42-49 lines. Aluminium is toxic to all systems of living organism. Due to its specific chemical properties, Al inhibits more than 200 biologically important functions and causes harmful effects. For instance, chronic exposure to Al has been proven to play a role in the development of neurodegenerative disorders: Parkinson’s dementia (4, 5, 6) or Alzheimer’s disease (7, 8, 9). The mechanisms of Al neurotoxicity are unclear, although research has shown that these mechanisms may mostly be attributed to the ability of Al to: (a) generate reactive oxygen species (ROS) or free radicals when metabolized, and to suppress the activity of antioxidant enzymes and other components of the antioxidant system in various organs, (b) impair signal transduction pathways in the cells, and (c) disturb calcium homeostasis. In addition, Al is one of the causes of hepatotoxicity (10, 11) and pathological processes in the testis, kidneys, and lungs (12, 13, 14). the brain was chosen for its importance in neurodegenerative diseases. Liver was chosen as known major site for detoxification.

50-58 lines. However, there is a range of chemical elements that play an important role in the antioxidant processes within cells (15, 16). One of such microelements is selenium (Se). The beneficial role of Se in human cells is due to its presence in at least 25 proteins – selenoproteins, part of which are directly involved in redox catalysis. Trace amounts of Se element are essential for cellular functions (17). Se is an element that reduces the risk of cancer (18), prevents cardiac diseases (19), and protects against the effects of ionizing radiation, heavy metals, and other toxic compounds. There is data indicating that Se strengthens the body’s immune system, thus reducing the risk of infection (20). In large amounts Se and its salts are toxic. Toxic doses of Se are able to negatively affect cellular redox status directly by oxidizing thiols, and indirectly by generating reactive oxygen species (ROS) (21).

We have corrected all values numbers in tables, setting the values on 2 digits. The experiments involved 8-10 mice, statistical calculations were based on analysis of repeated experiments.

Round 3

Reviewer 2 Report

This paper is appropriately revised along with the referees’ comments, so the manuscript can be published.

Author Response

Thank you very much for your time in evaluating our manuscript and for your valuable comments.

This manuscript is a resubmission of an earlier submission. The following is a list of the peer review reports and author responses from that submission.

Round 1

Reviewer 1 Report

The manuscript by Sadauskiene et al. presents the study on antioxidant enzyme activity (SOD, CAT and GR) in mice exposed to aluminium or selenium.

The figures need to be changed and please use standard deviation instead of SEM. I know that SEM is smaller than SD and looks better but it does not represent the distribution but is related to estimation of obtained mean from samples in relations to the whole population. The problem of blood brain barrier should be analyzed since the Authors studied brains. Oxidative stress reactions are very rapid. Why did you choose 8 -weeks administration? It would be interesting to show also the acute phase. Saying „Selenium also increased GR activity – by, accordingly, 36% and 24% (also in the dose-dependent manner).” is difficult since you have only two doses. Please describe what was the idea of selecting those two ions? What situation in human exposure it could represent. The Authors only studied the activity of enzymes and there are no data on oxidative stress. The paper could be improved if they include any results of oxidative stress damage to cells.

Author Response

Thank you very much for your time spent for the evaluation of our manuscript. We took into consideration your valuable remarks when correcting it.

According to your suggestion, we replaced Mean ± SEM with Mean ± SD, showing the change in the Methods section and providing respective values in tables that we replaced the figures with according to another Reviewer’s suggestion.

Why such doses of Al and 8 weeks of exposure? We had already analyzed the effect of acute (2-, 8-, and 24-hour as well as 14-day) exposure to various doses of Al on the level of oxidative stress and the antioxidant system in the blood, brain, and liver of mice after intra-abdominal injection of Al salt solution. The selected route of the administration of Al ensures a rapid entry of Al into the bloodstream and is suitable for acute exposure experiments. In humans, such route of the administration of larger amounts of Al is used in vaccination where Al is employed to improve the immunogenicity of vaccines. Recently, there has been a marked increase in the population’s fear of vaccination due to the presence of Al in the composition of vaccines, as this metal is associated with certain neural pathologies.

Al is also associated with Alzheimer’s disease, Parkinson’s disease, and other neurodegenerative disorders. A number of studies performed with experimental animals have demonstrated changes in the cognitive functions and morphological peculiarities of the CNS following the consumption of water with elevated Al concentrations. For this reason, we became interested in an experimental model with mice that were administered water containing an Al salt, since this is a natural route of entry of Al into the human body (Al enters the body with potable water, food, and medicines (e.g. antacids or buffered aspirin). Despite the fact that only up to 1.0% of the consumed amount of Al is absorbed in the digestive system, a long-term negative effect of Al cannot be ruled out completely even if lower concentrations of Al enter the body with potable water. For this reason, the aim of this study was to evaluate the long-term (2 months) effect of two different doses of orally administered Al. Another reason for our interest in the long-term effect was that we also perform histopathological examinations of the tissues and evaluate the effect of these elements on the increase in the body and organ mass of the mice (data not yet published).

Why Se was selected? Se is an element that can act as an antioxidant by increasing the capacity of the intracellular antioxidant system. On the other hand, at toxic doses, Se has the opposite effect, as it affects the cell’s redox status and may indirectly stimulate ROS formation – i.e. may act as a pro-oxidant. In acute exposure experiments, we had already evaluated the effect of Se on the level of oxidative stress and the antioxidant system in the blood, brain, and liver of mice after intra-abdominal injection of a Se salt solution. We had also evaluated the protective effect of Se after the injection of a combined Al + Se solution. For this reason, as we moved to long-term experiments, we sought to select several different doses of Se and to evaluate their effect on the capacity of the cellular antioxidant system (the results of this study). In addition, we also evaluated the effect of two doses of Se on the level of oxidative stress and determined Se concentrations in the brain (data not yet published). A further aim of our study would to evaluate the combined effect of Al and Se and, possibly, a protective effect of Se (this would complement the recommendations on the expedience of using Se supplements and the importance of avoiding their overdose).

Reviewer 2 Report

This experimental study describes the effects of doses of aluminum and selenium salts on the antioxidant endogenous system in the brain and liver of mice. The authors report their results but the presentation lacks of thorough explanation of the experimental outcomes. The conclusions remain speculative. In addition, I think that this contribution is suitable for other journals mainly focusing on antioxidant defense systems and oxidative stress. Thus I do not recommend publication in Catalysts. Last, in the Introduction the authors make some confusion between element and compound, referring for example at lines 30-35 to Al and Se as compounds. This is a chemically severe mistake. 

Author Response

Thank you very much for your time spent for the evaluation of our manuscript. We took into consideration your valuable remarks when correcting it.

Concerning the mistake in lines 30-35 you mentioned. Yes, we do understand that Al and Se are elements – but the animals received them not in the form of atoms but rather as certain salts (AlCl3 and Na2SeO3) in potable water, and for this reason we used the term “compounds” in the text. There was a mistake in the style of the sentence – we have corrected it.

Reviewer 3 Report

Result of abstract (Line 21-25) was not clear. Text in introduction was not enough to describe the objective of this article. Please provide reference in text of Line 130-132 “The much weaker effect of Al on some enzymes of the antioxidant system (SOD and CAT) might have been due to the fact that because of its physical and chemical properties, this metal has a relatively difficult entry into the body and very slowly accumulates in organ cells”. The data of antioxidant enzyme in liver means short-term effect. Data in brain means long-term effect. The manuscript had no discussion on it. Analyses of CAT, SOD, GR had different meaning, text in discussion had not fully described such meanings. Please evaluate change the data of Figure into Table.

Author Response

Thank you very much for your time spent for the evaluation of our manuscript. According to your remarks, we changed the abstract and made certain adjustments in the Introduction section.

As you suggested, we agree that the results of the experiments are more clearly visible in tables than in figures, and made respective changes.

Aluminum is known to accumulate differently in various organs. In our previous study, we had evaluated the distribution of Al in some mouse organs during acute exposure experiments: Sadauskiene I, Liekis A, Stanevičiene I, Viezeliene D, Zekonis G, Baranauskiene D, Naginiene R. Post-exposure distribution of selenium and aluminum ions and their effects on superoxide dismutase activity in mouse brain. Molecular biology reports. Dordrecht: Springer. Volume 2018, p. 1-7.

In this case, we provided a different reference (34): Krewski D, Yokel RA, Nieboer E, et al. Human health risk assessment for aluminium, aluminium oxide, and aluminium hydroxide. J Toxicol Environ Health B Crit Rev. 2007;10:1-269: “Normal tissue aluminium concentrations are greater in lung (due to entrapment of particles from the environment) than bone than soft tissues. Approximately 60, 25, 10, 3 and 1% of the aluminium body burden is in the bone, lung, muscle, liver and brain, respectively. Higher concentrations are seen in uremia and higher still in dialysis encephalopathy.”

Round 2

Reviewer 1 Report

The study should be reanalyzed. If you suggest that Al ions are an important threat from vaccinces you should elaborate that issue in the manuscript. If you studied other time points of exposure please show it. Otherwise one can expect that there will be another manuscript regarding 2 weeks, another regarding 10 weeks exposure, etc 

Author Response

Comments and Suggestions for Authors

The study should be reanalyzed.

The study was reanalyzed carefully.

If you suggest that Al ions are an important threat from vaccinces you should elaborate that issue in the manuscript.

In this case the threat from vaccines is not important as we used different model. Al and Se were administrated through the drinking water. This is the reason why we did not discussed this issue in this manuscript.

If you studied other time points of exposure please show it. Otherwise one can expect that there will be another manuscript regarding 2 weeks, another regarding 10 weeks exposure, etc 

In this manuscript we do not discuss other time points. We studied 8 weeks time frame. Because our previous studies have shown that 8-week time frame produces the most evident results for expressing an activities of antioxidant enzymes.

Reviewer 2 Report

I appreciate that the authors amended the text where required, to avoid

misleading terminology.

anyway, I remain on the opinion that the manuscript is not suitable for this journal. It represents an interesting study but for a restricted community and the results at this stage remain still speculative.

Author Response

We would like to thank you for your opinion about this manuscript itself.